

# Hanging glacier monitoring with icequake repeaters and seismic coda wave interferometry: a case study of the Eiger hanging glacier

Małgorzata Chmiel[1], Fabian Walter[2], Lukas Preiswerk[1,3], Martin Funk[1], Lorenz Meier[4], and Florent Brenguier[5]

[1]Laboratory of Hydraulics, Hydrology and Glaciology, ETH Zürich, Zürich, Switzerland
[2]Swiss Federal Institute for Forest, Snow and Landscape Research, Zürich, Switzerland
[3]Forensic Technology & Discovery Services, Ernst & Young, Zürich, Switzerland
[4]Geoprevent AG, Technoparkstrasse 1, Zürich, Switzerland
[5]Univ. Grenoble Alpes, Institut des Sciences de la Terre, Grenoble, France

**Correspondence:** Małgorzata Chmiel (chmielm@ee.ethz.ch)

**Abstract.**

Driven by the force of gravity, hanging glacier instabilities can lead to catastrophic rupture events. Reliable forecasting remains a challenge as englacial damage leading to large-scale failure is masked from modern sensing technology focusing on the ice surface. The Eiger hanging glacier, located in the Swiss Alps, was intensely monitored between April and August 2016

before a moderate 15,000 m$^3$ break-off event from the ice cliff. Among different instruments, such as an automatic camera and interferometric radar, four 3-component seismometers were installed on the glacier. A single seismometer operated throughout the whole monitoring period. It recorded over 200,000 repeating icequakes showing strong englacial seismic coda waves. We propose a novel approach for hanging glacier monitoring by combining repeating icequake analysis, coda wave interferometry, and attenuation measurements. Our results show a seasonal 0.1% decrease in relative englacial seismic velocity $dv/v$ and an

increase in coda wave attenuation $Q_c^{-1}$ ($Q_c$ decreases from $\sim$50 to $\sim$30). Comparison of $dv/v$ and $Q_c$ with air temperature suggests that these changes are driven by a seasonal increase in the glacier's ice and firn pack temperature that might affect the top 20 m of the glacier. Diurnal cycles of $Q_c^{-1}$, repeating icequake activity, and the velocity of the glacier front shift from cosinusoidal to sinusoidal variations under the presence of meltwater. The proposed approach extends the monitoring of the hanging glacier beyond the ice surface and allows for a better understanding of the glacier's response to time-dependent

external forcing, which is an important step towards improved break-off forecasting systems.

## 1 Introduction

Hanging glaciers are high-altitude glaciers that are inherently unstable and might produce catastrophic break-off events (Faillettaz et al., 2015). These glaciers are often frozen to the bedrock, which allows them to locate on steep slopes and consequently detaching ice masses can cause ice avalanches. Large ice avalanches, even relatively rare, can pose severe hazards to humans,

settlements, and infrastructure worldwide (e.g., Faillettaz et al., 2015; Tian et al., 2017), including Switzerland (e.g., Faillettaz et al., 2011).



A timely warning and evacuation often remain the only solution to protect the population (Faillettaz et al., 2015). Good prediction results are achieved with remote measurements of glacial surface velocities. It has been shown that the velocity of an unstable mass follows a power law increase in time before a break-off event (Flotron, 1977; Röthlisberger, 1977; Pralong and Funk, 2006; Faillettaz et al., 2008). However, factors responsible for the destabilization of large ice masses result from a combination of glacier geometry, ice rheology, and damage evolution, as well as basal motion and cannot be revealed with measurements of ice surface velocities alone (Pralong and Funk, 2006). Consequently, monitoring hanging glacier's subsurface changes and dynamic processes will likely improve warning capabilities. To date, however, such observations are largely missing and theories on damage evolution, basal sliding, and external forcing mechanisms are difficult to test.

Direct measurements on steep, heavily crevassed and avalanche prone glaciers tend to be challenging, sparse, and often difficult to interpret. This is also true for seismological observations on hanging glaciers. Previous seismological research mostly focuses on counting icequake related to englacial fracturing and their relation to break-off events (e.g., Faillettaz et al., 2011; Preiswerk et al., 2016). Yet, seismic waves contain additional information about their sources and the medium, through which they travel which can be been exploited with repeating icequakes and seismic interferometry (e.g., Allstadt and Malone, 2014).

Repeating icequakes have been observed in Antarctica by Anandakrishnan and Bentley (1993); Smith (2006); Zoet et al. (2012), in Greenland (e.g., Roeoesli et al., 2016), beneath Alpine valley glaciers (e.g. Helmstetter et al., 2015; Walter et al., 2020; Gräff and Walter, 2021), and glacier-covered volcanoes [Thelen et al. (e.g. 2013); Allstadt and Malone (e.g. 2014)]. Most likely related to repeated slip over frictional asperities, repeating icequakes act as a repeating seismic source whose signals can be used to detect small changes in the medium over time with coda wave interferometry (e.g., Snieder et al., 2002; Sens-Schönfelder and Brenguier, 2019).

Coda wave interferometry is an approach based on seismic interferometry (Curtis et al., 2006) to monitor those subsurface structural variations which cause changes in seismic velocities. It exploits the later arriving, multiply scattered seismic signals ('coda') rather than the direct phases of a seismic event. Coda wave interferometry has detected subsurface seismic velocity variations induced by perturbations of crustal properties (Brenguier et al., 2008a; Niu et al., 2008), volcanic flank movement (Brenguier et al., 2008b; Obermann et al., 2013a; Sens-Schönfelder and Wegler, 2006), stress distribution in mines (Olivier et al., 2015), loading and unloading of glacial mass (Mordret et al., 2016), and changes in permafrost thickness (James et al., 2018). Studying changes in the envelope of the coda waves also allows for an estimation of seismic attenuation, another measure of the medium's elastic properties (Aki and Chouet, 1975). Yet, the applications of coda wave interferometry are limited on Alpine glaciers due to limited englacial scattering resulting in weak coda (Sergeant et al., 2020, e.g.,). However, pervasive fracturing within hanging glaciers and multiple lateral reflections within small glacial basins can potentially generate sufficient coda to use coda wave interferometry (Podolskiy and Walter, 2016).

Here, we propose a novel approach for hanging glacier monitoring by combining repeating icequake analysis and coda wave interferometry. For that, we use seismic data recorded by a 4-station network deployed on the Eiger hanging glacier in Switzerland between April and August, 2016. By investigating icequake occurrence we find over 200,000 repeating events showing strong englacial coda. We compile a catalog of 23 selected clusters by automatically searching the events over a single



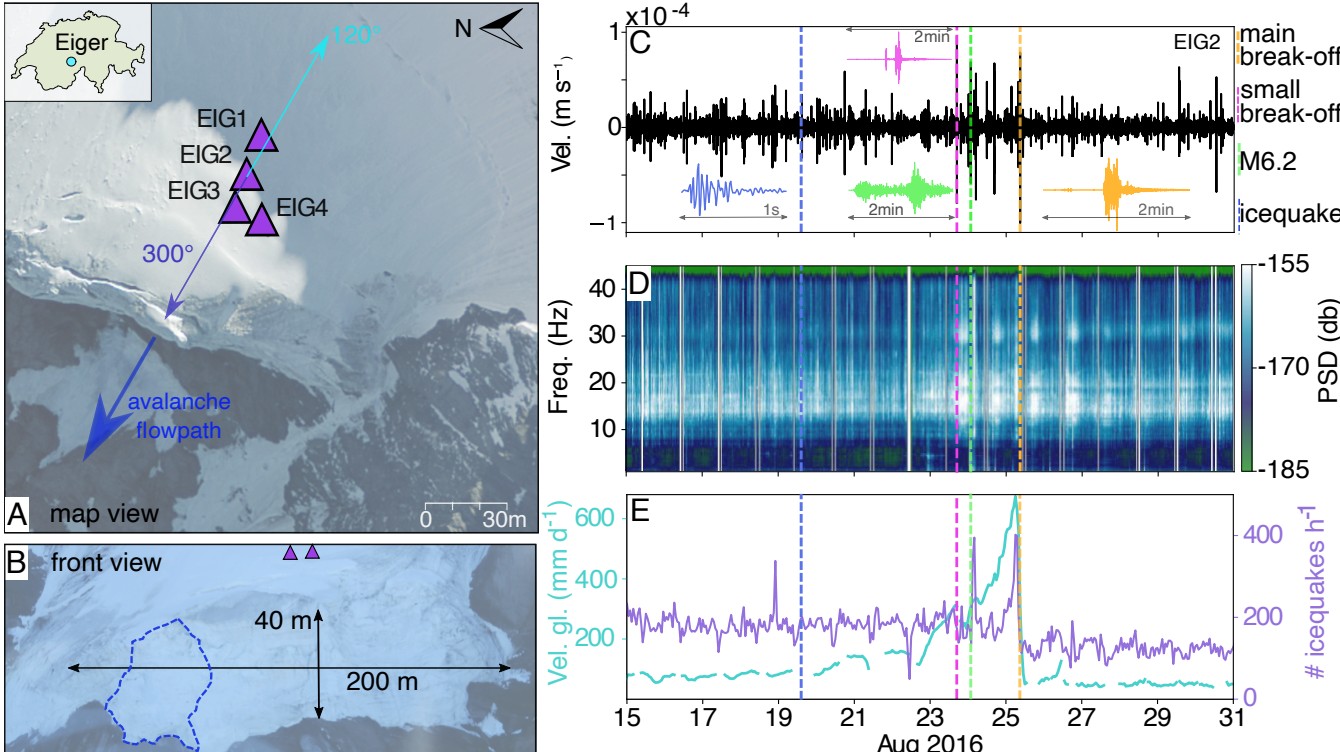

**Figure 1.** Study site: Eiger hanging glacier. (A) Ortophoto of the Eiger hanging glacier one day after the main break-off event. The position of four seismic stations installed at the glacier is marked with purple triangles. The dominant backazimuth (120°) of the repeating icequakes is marked with a light blue arrow together with its 180° uncertainty (dark blue arrow). The end of the dark-blue arrow points towards the lamella position that broke off. Photo source: swisstopo flight, line 1308201608260940, August 26, 2016. The inset shows the location of of the Mount Eiger in Switzerland. (B) Photo of the ice cliff (August 31, 2016). The approximate spatial extension of the broken-off ice lamella is marked in dashed blue line. (C) Seismic signals recorded on the vertical component of the EIG2 station 10 days before and 5 days after the main break-off with a corresponding spectrogram in (D). (E) Evolution of the maximum velocity of the glacier front (measured using an interferometric radar) smoothed with a 1h-moving average and an hourly icequake occurrence rate. The time occurrences of a single icequake, small and main break-offs, Amatrice earthquake (M 6.2) are represented in dashed vertical bars and colors indicated in Figure 1C.

3-component station operating throughout the whole acquisition period. We use the clusters as a repetitive source required for (1) coda wave interferometry to monitor englacial seismic velocity changes, and (2) coda wave measurements of seismic attenuation. Our results show that the proposed approach is suitable for monitoring diurnal and seasonal changes in elastic
properties of a hanging glacier.



## 2  Study site

The Eiger hanging glacier, located on the west face of the Eiger mountain, Switzerland, extends from 3,500 to 3,200 m a.s.l. with a surface slope of 20° at the terminus [Margreth et al. (2017), Figure 1A-B, B1]. The surface area of the Eiger hanging glacier was 0.08 km$^2$ in 2016 (Huss et al., 2013) with a mean and maximum thickness of about 40 m and 70 m, respectively. Lüthi and Funk (1997) determined the average glacier flow velocity as 7 m a$^{-1}$ with englacial temperatures ranging between -5° to 0°. The Eiger hanging glacier is polythermal, meaning that water coexists with glacier ice at the glacier base, except the base of the frontal part which is cold (entirely frozen to the bed) (Lüthi and Funk, 1997). The glacier lies almost entirely in the accumulation zone where it exhibits a positive annual net surface balance. The glacier extent is limited by a topographic bedrock step, which leads to a steep ice cliff from which break-off events occur periodically. Typical volumes of unstable ice lamella that break off are $< 10,000 \, \text{m}^3$. The consequent ice avalanches are large enough to endanger hiking paths, ski infrastructure, and the train line that leads to Junfraujoch (one of Europe major tourist destinations, see Figure B1 for an overview of the glacier location). To warn against the break-off events, a monitoring system has been installed next to the glacier since 2016, which includes an automatic camera (two photos per day of the glacier front) and an interferometric radar measuring the velocity of the glacier front (Meier et al., 2016). Velocities of the unstable ice lamella increase as a power law function of time prior to failure and can be used for forecasting [Pralong and Funk (2006), Figure 1E].

In April 2016, a significant crevasse was observed behind the glacier front, indicating an impending break-off event. We installed four 3C Lennartz seismometers (natural frequency 1 Hz) on the glacier between April and August 2016 to monitor its seismic activity [see Preiswerk (2018) for details on acquisition]. Avalanches, snow fall, and other factors associated with high altitude conditions strongly challenge instrument maintenance. However, one station recorded continuously for 4.5 months (EIG2), and up to three seismic stations operated simultaneously (Figure 1A).

In the morning on August 25, 2016 an ice mass of 15,000 m$^3$ broke off the hanging glacier (Figure B2). This break-off event was the largest since 1991 (Margreth et al., 2017). The ice avalanche missed a train station and came to rest 1200 m vertically below the glacier. Seismic stations recorded the main break-off event together with precursory break-off events (e.g., a small break-off on August 23), and abundant icequake activity prior to the break-off events (Figure 1C-E).

## 3  Methods

### 3.1  Seismic activity on the glacier and repeating icequakes

We first investigate local icequake occurrence recorded by the seismic array before, during, and after the break-off event (Figure 1E). Following Preiswerk et al. (2016) we determine the icequake activity using a short time average/long time average trigger [Allen (1978); see Appendix A1 and Preiswerk (2018) for details].

Next, we extend the analysis to repeating icequakes. The repeating events imply sources in close proximity with the same source mechanism, resulting in highly similar waveforms (Poupinet et al., 1984). They can be easily and comprehensively searched for using cross-correlations and such methods have frequently been used to detect icequakes (e.g., Mikesell et al.,



2012; Thelen et al., 2013; Helmstetter et al., 2015). Due to the high number of events, an automated procedure reducing manual waveform inspection is necessary. For that, we perform a two-step, automated, and cross-correlation based analysis. We first

use RedPy [Repeating Earthquake Detector in Python, Hotovec-Ellis and Jeffries (2016)] that automatically detects repeating icequakes in continuous data. RedPy requires no template wavefrom and repeating icequakes are clustered based on cross-correlation across multiple stations (Hotovec-Ellis and Jeffries, 2016). We run automatic repeating icequake detection with RedPy using data from two stations: EIG2 and EIG4, between August 11 and 31, 2016. The data are high-pass filtered at 1 Hz to focus on high frequency signals related to the glacier dynamics (Podolskiy and Walter, 2016).

We find thousands of repeating events with a correlation coefficient threshold of 0.9. In order to complete the analysis over the entire monitoring period, we manually choose 23 clusters showing strong coda. We construct template waveforms by stacking the icequake waveforms within each cluster. Next, we systematically search for similar icequakes with template matching by scanning continuous data recorded at the EIG2 station (e.g., Gibbons and Ringdal, 2006). For the implementation, we use the Fast Match Filter [FMF, Beaucé et al. (2018)] and we set the correlation coefficient threshold to 0.5 to provide a

complete catalog.

We also determine a backazimuth of the repeating icequakes using a polarization analysis (e.g., Jackson et al., 1991; Greenhalgh et al., 2018). The polarization analysis is based on a singular value decomposition of a complex covariance matrix calculated over 3-component windows of repeating icequake waveforms. The real part of the principal eigenvector is used to estimate the backazimuth direction assuming that the first-arrival is generated by linearly polarized P-waves (Greenhalgh et al.,

2018). The backazimuths of clusters are obtained for a range of 0-180° as there is a 180° ambiguity in the values. The details of the repeating icequake detection with RedPy and FMF, and back-azimuth analysis can be found in Appendix A2 and A3.

### 3.2   Coda wave: interferometry and attenuation

We then use the 3-component waveforms from the 23 clusters as repetitive sources to investigate elastic changes within the glacier. We calculate relative velocity variations ($dv/v$) using the doublet method, also called Moving Window Cross Spectral

technique (Poupinet et al., 1984; Fréchet et al., 1989; Clarke et al., 2011; Lecocq et al., 2014). The time differences ($dt$), between an analysed trace and a reference constructed from the stack of all the icequake signals, are calculated at a given time lag ($t$) through cross-correlations. This allows to assess the $dt/t$ value through the slope (Figure A7), and then, using the relationship for a homogeneous velocity change in a medium (Snieder et al., 2019) we obtain: $dv/v = -dt/t$. Given the dominant frequency band of repeating icequake signals (Figure 2D), we perform the analysis in the 10-40 Hz frequency band.

We use a 1-s-long time window starting at 0.5 s, after the direct arrivals. The icequake signals are averaged over a 3-day (seasonal variations) and 4-hour (diurnal variations) moving window (step=1h) to stabilize the measurement. To measure the diurnal variations, we stack a maximum of 3 events per hour with the highest cross-correlation coefficient obtained from the template matching. We also estimate changes in the coda wave attenuation ($Q_c^{-1}$) by measuring the decay slope of the coda envelope on the basis of Aki and Chouet (1975) model. Finally, we average 69 measurements (23 clusters times 3-components)

to obtain the final results. Technical details of these methods can be found in Appendix A4.

**Figure 2.** Repeating icequake analysis. (A) Timeline showing daily activity of repeating icequake clusters. Each daily icequake occurrence is a circle plotted on a line corresponding to its cluster. The color of each circle corresponds to the daily average of cross correlation coefficients between individual events and the termplate. The cluster name is labeled at right. The number of icequakes in each cluster is represented with a histogram (between 1,574 events for cluster 10 and 20,292 events for cluster 1, events that occurred before and during the melt season are marked in dark and light blue, respectively). The air temperature is marked in blue. (B) Interevent time of cluster 0. (C) Evolution of daily icequake signals recorded at the vertical component of station EIG 2 for cluster 0. Normalized average of the icequake signals is showed in blue. (D) Spectrogram of the icequake signals in (C). Time (0.5-1.5) s and frequency (10-40) Hz windows used for CWI are marked with black arrows.





## 4 Results

### 4.1 Seismic activity on the glacier

Figure 1E presents the rate of seismicity before and after the break-off event (in purple) compared with the maximum velocity of the glacier front (in blue). Long-term changes in event occurrence are presented in Figure A1. The glacier front significantly accelerates on August 23 (mean measured velocity > 20 cm day$^{-1}$ compared to 4-5 cm day$^{-1}$ before the break-off events, Figure S3). After the main break-off (August 25) the velocity drops below 10 cm day$^{-1}$. Faillettaz et al. (2011) showed that on another hanging glacier located in the Swiss Alps, Weisshorn glacier, the icequake activity accelerated together with the glacier front displacement ~5 days prior to the failure of an unstable large ice mass (volume ~120,000 m$^3$). We observe an increase in the seismicity rate only ~6 hours before the main break-off event (up to ~400 events h$^{-1}$) and drop to a lower level after the break-off (100 events h$^{-1}$). However, before the small precursory break-off on 23 August, seismicity did not clearly increase. On the other hand, our results show elevated seismicity two hours after the passing of the teleseismic waves of M 6.2 Amatrice earthquake (e.g., Chiaraluce et al., 2017), around 01:00 UTC on August 24. Moreover, another peak in seismic activity is visible on August 18. After a closer examination of seismic activity on the Eiger hanging glacier, we observe recurrent 1-2 hour long bursts of seismic activity that become ~10 times more frequent in melt season.

The difference in icequakes activity at Weisshorn and the Eiger hanging glacier might be related to different geometries, thermal regime at the glacier bed, type of instability, and volume of break-off events. The Eiger hanging glacier is polythermal and is located on a terrace, while Weisshorn hanging glacier is cold (entirely frozen to its bedrock) and rests on a steep slope (Faillettaz et al., 2015). For entirely frozen glaciers and for break-off events caused by an instability of a large glacier slab, the maturation of the rupture is associated with a typical time evolution of surface velocities and passive seismic activity (Pralong and Funk, 2006). However, for polythermal glaciers, such as the Eiger hanging glacier, no clear and easily detectable seismic precursors are known. Our results confirm that icequake occurrences alone are not always suitable for early-warning purposes. However, further insights can be gained by analyzing individual icequake signals.

### 4.2 Analysis of repeating events

Figure 2A-B shows repeating icequake activity compared with air temperature. The air temperature was measured at MeteoSchweiz weather station Jungfraujoch and is corrected by +1° C to the altitude of Eiger hanging glacier, 3 km away. The amount of surface melt is proportional to the cumulative temperature above the melting point of 0° C over a given period (e.g., Wake and Marshall, 2015). All the repeaters showed an increased activity after June 21, 2016 when the air temperature exceeds 0° C (gray dashed line in Figure 2A-B) and surface melt takes place at the glacier.

For most clusters, the cross-correlation coefficient between the template and the icequake signals increases from 0.5 to >0.7 within the melt season. This progressive increase in the cross-correlation coefficient can be caused by an improved similarity of the icequake waveforms to the template which is constructed using icequake waveforms from August 11-31. However, the cross-correlation coefficient drops for clusters 4, 7, and 11 just after the main break-off event. We discuss this in the next section. With the increasing number of events the interevent times decrease during the melt season (from > ~12 h before to ~20 min





during the melt season). Figure 2B shows a scatter plot of the interevent times as a function of time for cluster 0. The activity
of this cluster is strongly correlated with melt: the intervent times strongly decrease (by two orders of magnitude) when air
temperature increases to above 0° C. For a different representation on the evolution of number of events and cross-correlation
coefficient the reader can refer to Figure A2.

Most of the clusters (12 out of 23) show dominant directions varying between 90° and 150° (Figure A4). The dominant
azimuth at 120° and its 180° uncertainty, indicate the origin of most clusters either from the back of the glacier where a large
crevasse is visible and where glacier is not frozen to the bed, or from the unstable glacier front.

Seismograms of cluster 0 (Figure 2C) show strong phase coherence of direct arrivals and pronounced coda. The coda is
probably generated by waves reflected at the ice/bed interface of the glacier and by waves scattered at the pervasive fracturing
within the hanging glacier. The incremental changes in the coda can be driven by the perturbation in seismic velocities, scat-
terers, and source position (Snieder et al., 2002). Considering moderate glacier flow velocity [7 m a$^{-1}$, Lüthi and Funk (1997)],
extended life-cycle of large crevasses compared to the duration of the monitoring period (Colgan et al., 2016), high sensitivity
of seismic velocities to changes in ice elasticity (Röthlisberger, 1972), and finally the minimum wavelength we work with
(~40 m for surface waves at 40 Hz), we hypothesize that changes in the coda are driven by seasonal changes in glacier elastic
properties. A discussion of the implications and related uncertainties of our hypothesis is given in Appendix A5.

### 4.3 Relative seismic velocity and attenuation variations

Coda-wave interferometry results show moderate variations of +/-0.25 (%) in relative englacial seismic velocities. From mid-
April to mid-July we observe a long-term decrease in $dv/v$ (~0.1% peak-to-peak amplitude) that inversly correlates with
the long-term increase in the air temperature (Figure 3A-B). Coda wave attenuation ($Q_c^{-1}$) increases during the monitoring
period. The seismic quality factor $Q_c$ varies from ~50 to $Q_c$ =~30 before and during the melt period, respectively, indicating
very high attenuation of seismic waves in the glacier (Figure 3D). The attenuation in the Eiger hanging glacier can be further
enhanced by the scattering loss at crevasses. The $Q_c$ values are similar to previous attenuation estimations for glacial studies
with surface waves [Q = 35 at 20 Hz, Jones et al. (2013)) and body waves (Q=20 at 30-500 Hz Helmstetter et al. (2015)].
However, to our knowledge this is the first study providing estimates on $Q_c$ in the ice.

The $dv/v$ and $Q_c^{-1}$ measurements remain overall stable from the beginning of July to mid-August with short-time variations.
For example, the drop in $dv/v$ around July 10 (marked in Figure 3A-B with gray arrows) can be related to englacial damage due
to rapid refreezing of meltwater near the surface that leads to a volumetric expansion. Starting from the beginning of August
the glacier front starts to accelerate. We also observe an increase in $dv/v$, followed by a decrease in attenuation $Q_c^{-1}$. These
observations lack straightforward dependence on the temperature and we discuss a possible explanation in the next section.

### 4.4 Diurnal cycles

To better understand and further quantify changes occurring at the hanging glacier we now focus on daily cycles. We average
the measurements of $dv/v$, air temperature, the maximum velocity of the glacier front, attenuation $Q_c^{-1}$, and the normalized
number of repeating icequakes (repeaters) per cluster in hourly bins. We define four distinct time periods: (1) April 15 -June 21,

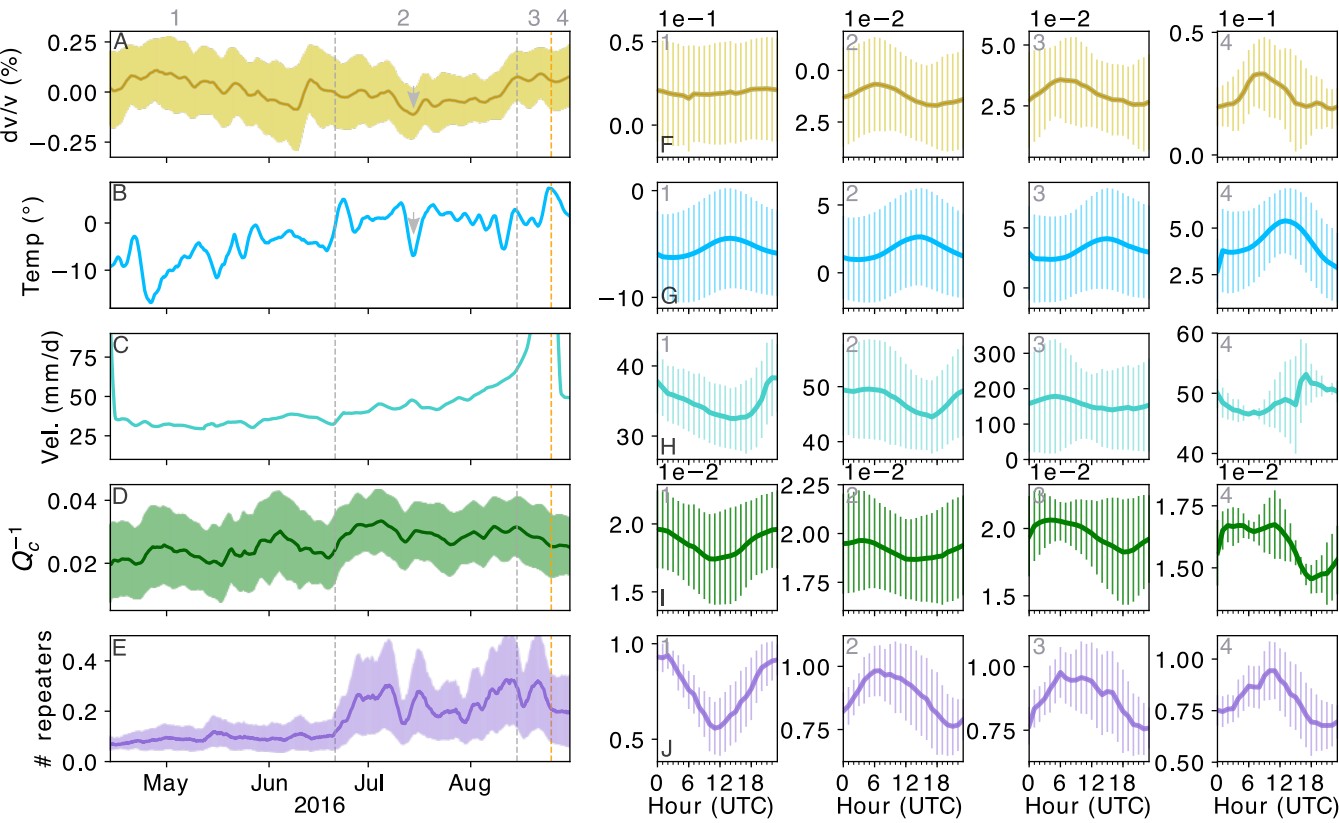

**Figure 3.** Results of coda wave interferometry (CWI). (A) Averaged $dv/v$ results and its standard deviation, (B) air temperature, (C) zoom on the velocity of the glacier front, (D) Averaged coda attenuation $Q_c^{-1}$ and its standard deviation, (E) Averaged normalized number of repeating icequakes (repeaters) and its standard deviation. All measurements are smoothed with a 3-day moving average. Average diurnal cycles and their standard deviations for four different time periods, from the top: (F) $dv/v$, (G) air temperature, (H) velocity of the glacier front, (I) coda attenuation $Q_c^{-1}$, and (J) number of repeaters. The measurements are smoothed with a 8-hour moving average prior to averaging.

(2) June 21-August 15, (3) August 15-27, and (4) August 27-31. Gray dashed lines in Figure 3A-E delimit these time periods, and the orange dashed line marks the main break-off event. We summarize our observations in the following:

1. April 14–June 21: the daily temperatures remain below 0° C. The glacier surface temperature is increasing with increasing air temperature. We observe limited repeater activity with a recurrent presence of 5 clusters. The velocity of the glacier front is the fastest at night ($\sim 40$ mm d$^{-1}$) with diurnal variations of $\sim 5$ mm d$^{-1}$ ; the attenuation $Q_c^{-1}$ ($\Delta_{diurnal}Q_c{=}\sim 5$) and the repeating icequake activity also peak at night hours. The $dv/v$ show no distinct diurnal cycle before the melt season, in contrast to $Q_c^{-1}$.

2. 20 June-15 August: the daily temperatures remain above 0° C and surface melt takes places at the glacier. The phase of diurnal cycles ($Q_c^{-1}$, velocity of the glacier front, and repeater activity) shifts by $\sim 6$ h from cosinusoidal to sinusoidal





functions. $dv/v$ ($\Delta_{diurnal}dv/v$=~0.01%) and $Q_c^{-1}$ variation follow a similar sinusoidal trend that inversely correlates with temperature. The velocity of the glacier front increases by 10 mm d[-1] and we observe an enhanced repeater activity with an appearance of 18 additional clusters. The velocity of the glacier front and the repeating icequake activity peak in the morning hours.

3. August 15-27: the glacier front strongly accelerates with observed values of $>$100 mm d[-1] and diurnal variations of ~40 mm d[-1]. The pronounced activity at the glacier front is clearly indicated on the camera photos by the opening of several crevices in the approximately 10 m above the front and the precursory break-off activity starts on August 14. $dv/v$, $Q_c^{-1}$, and the repeater activity show the same pattern of diurnal variations as in (2).

4. August 27-31: after the break off events, the glacier velocity is strongly reduced ($<$60 mm d[-1]) with the highest val-
ues in the afternoon. Diurnal variations of $dv/v$ and $Q_c^{-1}$ become more pronounced ($\Delta_{diurnal}dv/v$=~0.015% and $\Delta_{diurnal}Q_c^{-1}$=~10) with the elevated air temperature.

## 5   Discussion

The spatial sensitivity of the coda waves at the Eiger hanging glacier is uncertain due to unknown source positions, unknown contribution of surface and body waves to the englacial coda, and 3D-wavefield effects caused by the glacier's distinct geometry
(Preiswerk et al., 2018). Yet, considering strong spatial sensitivity of coda waves to source-receiver locations and assuming surface-wave like depth sensitivity (Obermann et al., 2013b), the $dv/v$ and $Q_c^{-1}$ measurements should allow to probe at least the top ~40 m of the glacier [the depth sampled is one-third of the wavelength used, Gazetas (1982)]. Moreover, the backazimuth analysis indicates that the repeaters originate from various directions assuring lateral sensitivity to englacial changes.

A limited number of studies analysed seismic attenuation and velocity variations in glacier ice, but most of them shown
that seismic velocity and $Q$ usually inversely correlate with ice temperature (e.g., Spetzler and Anderson, 1968; Röthlisberger, 1972; Peters et al., 2012). This is in agreement with the seasonal $dv/v$ and $Q_c^{-1}$ variations that we observe. The comparison to air temperature and inferred coda wave spatial sensitivity suggest that the long-term decrease in relative seismic velocities is induced by seasonal variations in the ice and firn pack temperature that affect the glacier's top ~20 m (Sanderson, 1978). The seasonal temperature variations near the glacier surface are also reflected in the increase of seismic attenuation that is highly
sensitive to temperature (Peters et al., 2012).

Diurnal cycle in $dv/v$ can be partially induced by varying melt water content in crevasses and smaller fractures at the glacier surface, as even a small volume fraction of water has a large impact on seismic velocities (Spetzler and Anderson, 1968). Lack of diurnal variations before the melt season might indicate that the attenuation has a higher sensitivity to small changes in the subsurface properties than seismic velocities, which was already shown in crustal seismology (Töksoz et al., 1979).
Interestingly, diurnal $Q_c^{-1}$ cycles inversely correlate with the air temperature during the melt season. This could be possibly related to other mechanisms influencing glacier elastic properties, such as small crack formation near the glacier surface due





to rapid refreezing of melt water. However, more studies are needed for a sound understanding of the observed $dv/v$ and $Q_c^{-1}$ changes.

For melt periods (2) and (3) the velocity of the glacier front is the highest in the morning hours. This is rather surprising as
elevated glacier velocities are more typical in the afternoon hours as a result of meltwater enhanced sliding (e.g., Kamb et al., 1994; Hewitt, 2013; Liu et al., 2019). Such a trend, however, can be observed for the last period (4) after the lamella broke from the ice cliff. In this period, the glacier front velocity measurements might better represent the overall bulk-glacier flow.

If we assume that the variations in the velocity of the glacier front are representative of the overall glacier flow variations, the change in diurnal cycles between the periods (2-3) and (4) hints towards a time shift between the maximum velocity of the
stable and unstable part of the glacier. Such a time shift can suggest a time delay between the transfer of stresses generated in the back of the glacier to the front of the glacier that is frozen to its bed, and the unstable part. This behaviour could be potentially described using a viscoelastic model based on, e.g., a spring (elasticity)-dashpot (viscosity) Maxwellian model (e.g., Podolskiy et al., 2019). The lamella could be described as an additional dashpot introducing a time shift between the applied stress and the deformation at the glacier front. This is however beyond the scope of this paper.

Finally, the diurnal variations in the velocity of the glacier front could also be caused by atmospheric effects that are different at night and during the day (e.g., Luzi et al., 2004). Atmospheric effects might be the same order of magnitude as surface velocities causing a strong impact on the measured surface deformations. In the future studies, the diurnal variations of surface velocities measured with the interferometric radar could be verified with an independent method, for example a differential GPS sensor placed at the glacier front (e.g., Preiswerk et al., 2015).

The break-off events correlate in time with the maximum air temperature (Figure 3) and there were successfully forecasted with the remote measurements of glacier surface velocities. The increased amount of meltwater filling the front crevasse can directly accelerate crack propagation to the entire ice thickness (Faillettaz et al., 2011). But, in addition, meltwater penetrates through cracks down to the glacier bed in the stable part. The increased subglacial water pressure promotes basal sliding, which in turn increases the shear stresses at the front (Lüthi and Funk, 1997). This stress accumulation could possibly explain the
$dv/v$ increase (e.g., Nur and Simmons, 1969) and $Q_c$ decrease due to microcrack closure in the stable part of the glacier (e.g., Töksöz et al., 1979) prior to break-offs. Yet, whether or not the stress accumulation consecutively makes the ice mass unstable, also depends on how strongly it is supported at the lateral edges; this is not known yet.

### 5.1 Repeating icequake interpretation

Icequakes can arise from surface crevassing, hydraulic fracturing, opening and closing of tensile faults, and glacier stick-slip
movement Podolskiy and Walter (2016). The tendency of icequakes to form clusters of thousands of events and high waveform similarities (Figure 2C) all suggest repeated source (e.g., shear faulting), rather than irreversible fracturing process. We detect ∼10 times more repeating icequakes during the melt season, which indicates an influence of meltwater and subglacial water pressure on their activity. Clusters 11, 17, and 18 show a decreasing activity after the break-off events. If we consider only repeaters with cross-correlation coefficients > 0.7, the activity of cluster 18 stops completely after the break-off (Figure B3) and



this is a good indicator of the cluster origin from the unstable glacier front due to crevasse opening. Moreover, the backazimuth analysis points towards the glacier front (300° with a 180° ambiguity).

Using the threshold of 0.5 in the template detection with a single station provides us with a complete catalog that we need for coda wave measurements. However, it makes the interpretation of individual icequake detections more uncertain. To resolve the origin and the source mechanism of the repeaters, an enhanced sensor coverage would be needed that could be obtained with, for example, a distributed-acoustic sensing (DAS) system deployed at the glacier surface (Walter et al., 2020). Localized repeaters originating from the glacier bed would allow to monitor changes at the glacier-bed interface.

## 6 Conclusions

Seismic measurements on hanging glaciers, even though technically challenging, give unique opportunity to probe the glacier interior with icequake repeaters and coda waves. Surface glacier velocity measurements allow for a timely prediction of break-off events, although they cannot reveal complex englacial changes taking place within polythermal hanging glaciers. For a better forecasting of catastrophic break-off events, we need a better understanding of the glacier's response to time dependent external forcing. Our results show the influence of air temperature and meltwater on glacier elastic properties and dynamics, that are reflected through the seasonal increase in repeating icequake activity, decrease in relative seismic velocity, and increase in coda attenuation. Our approach allows to extend the monitoring of hanging glaciers beyond the ice cliff. Permanent seismometer installations in the proximity of the glacier could be sufficient to perform the proposed repeating icequake and coda-based monitoring in the long term. Changes in coda attenuation and relative seismic velocities could provide an indication on the glacier's thermal state. This understanding could be used to detect warming or cooling trends at the ice/bed interface that could affect the glacier flow and its stability, and to contribute towards improved forecasting systems.

*Code and data availability.* Obspy Python routines (www.obspy.org) were used to download waveforms and pre-process seismic data. REDPY can be downloaded from: https://github.com/ahotovec/REDPy, and FMF from https://github.com/beridel/fast_matched_filter. The data from the EIG network is collected under the network code 4D (https://doi.org/10.12686/sed/networks/4D) and all seismic data available on request in the archives in the Swiss Seismological Service, http://www.seismo.ethz.ch/en/research-and-teaching/products-software/fdsn-web-services/ and the European Integrated Data Archive (EIDA), http://www.orfeus-eu.org/data/eida/. Interferometric radar data supporting this research have restricted availability at https://data.geopraevent.ch/index.php. To gain access, please contact Lorenz Meier.

## Appendix A: Methods

### A1  Seismic activity analysis

We use the short time average (STA) long time average (LTA) algorithm Allen (e.g., 1978) to evaluate the icequake activity at the Eiger hanging glacier. The STA/LTA algorithm continuously averages the absolute amplitude of a seismic signal in





two consecutive moving-time windows. Long time window (LTA) is sensitive to changing seismic noise and the short time
window (STA) provides information about seismic events. When the ratio of both exceeds a pre-set value, an event is declared.
Following Preiswerk (2018), we combine two sets of parameters sta=0.2 s and lta= 5s, sta=0.08 s and lta=0.8 s to detect different
types of events, and thresholds 5 and 2 for trigger on and off. We use coincidence triggering to avoid local artefacts, meaning
that the event has to be recorded on 4 channels (2 stations). We also account for changing trigger sensitivity by leaving out
low amplitude events (Walter et al., 2008), with median amplitude of events <2e-7 m s$^{-1}$. We do so, because the sensitivity of
detection changes diurnally and seasonally depending on the number of working stations and different noise levels when the
surface melt starts on the glacier. Daily icequake occurrence rate for the entire monitoring period (April 14-August 31, 2016) is
presented in Figure A1 together with the maximum velocity of the glacier front, and daily temperatures. For additional details
of icequake activity on the Eiger glacier, we refer the reader to (Preiswerk, 2018).

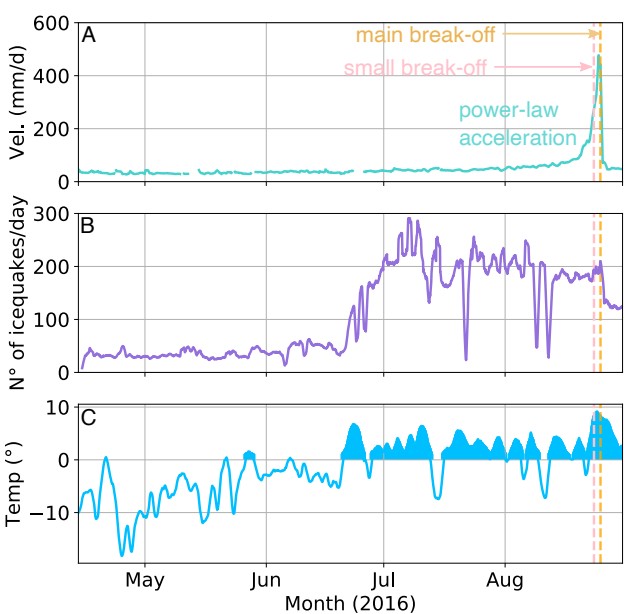

**Figure A1.** (a) Evolution of the maximum velocity of the glacier front measured from April 15 to August 31, 2016 using an interferometric
radar. The maturation of the rupture is associated with a power-law acceleration of the glacier front velocity. (b) Icequake occurrence rate:
results of the STA/LTA detection algorithm. (c) Air temperature recorded at Jungfraujoch (days with positive temperatures are filled in blue).
All measurements are smoothed with a 24h-moving average. The vertical dotted line indicate small precursory break-off events (in pink),
and the main break-off event (in orange).

## A2   Repeating icequakes

1. Repeating icequakes: RedPy


We use RedPy (Hotovec-Ellis and Jeffries, 2016) to investigate the occurrence of repeating icequakes between August 12 and 31, 2016. The RedPy detector runs on seismic data recorded at stations EIG2 and EIG4 (the data are high-pass filtered at 1 Hz). RedPy first runs an STA/LTA triggering algorithm and then a clustering algorithm based on cross-correlations. A cluster contains all events that correlate with at least one other event in the cluster above the correlation threshold. Clusters can combine if a new event correlates with an event in two or more clusters. We use similar settings in RedPY as for the STA/LTA: 4 channels need to be triggered in STA/LTA, we use 6 channels in total (2 stations), lta= 5, sta= 0.2, trigon= 5, trigoff= 2. The cross-correlation threshold of 0.9 has to be exceed on 4 channels for an event to be counted.

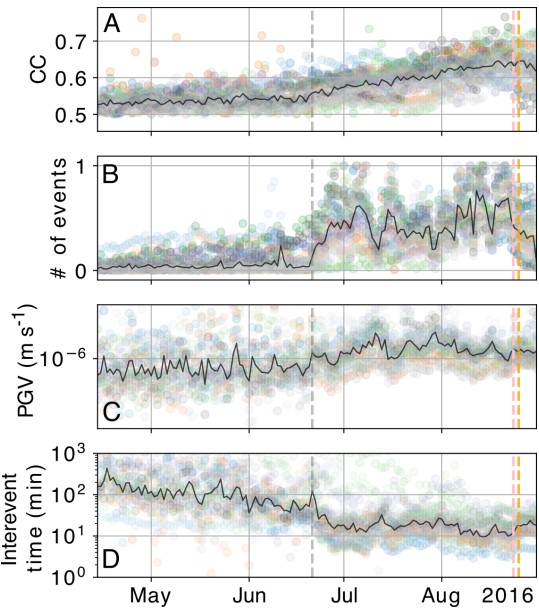

**Figure A2.** Evolution in daily repeater activity. Each cluster is represented in different color and median values calculated over all the cluster are presented in black lines.(A) Cross-correlation coefficient between between individual events and the stack of all the events in the cluster, (B) Normalized icequake occurrence in individual clusters, (C) Peak-ground velocity (PGV), and (D) interevent time.

2. Template matching

We perform a template matching over the entire monitoring period to complete the repeater catalogs. We first define templates by stacking icequake signals per cluster. The templates are cross-correlated with continuous signals recorded at the EIG2 station. This is the only station that was operational for the entire monitoring period. For the implementation, we use Fast Matched Filter [FMF, (Beaucé et al., 2018)]. FMF first computes normalized cross-correlations between a template and a sliding time window for each signal component and then the average correlations over the three components. We consider a correlation peak as a potential detection if the correlation exceeds 0.5. FMF returns the time series of the average correlation coefficients (CC) calculated for each template. The sliding time windows are taking every



sample. We group the detections within the time window of +/-1.5 template length and then we keep only the highest correlation coefficients to avoid double detections. Moreover, we cross-check detections for all clusters to eliminate double detections, by keeping the detections in the cluster with the highest cross-correlation coefficient. Figure A2 shows evolution in daily repeating icequake activity. Figure A3 shows vertical ground velocities from individual icequakes, their stacks, and amplitude spectra from the stack for each cluster.

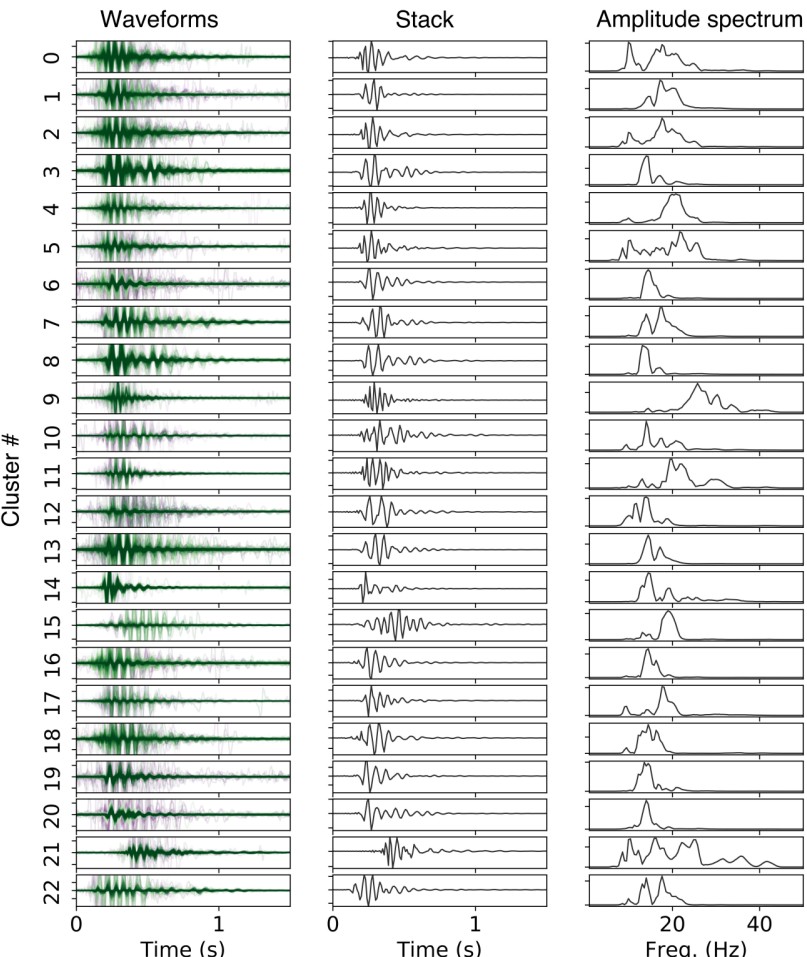

**Figure A3.** (left) Spaghetti plot of vertical ground velocities from individual icequakes in each cluster, and their stack (middle). Stacking procedure suppresses the noise and enhances lower frequencies. (right) Stack amplitude spectrum.




## A3 Backazimuth analysis

We also determine the dominant icequake backazimuth by using the previously defined templates. We use an approach based
on a singular value decomposition (SVD) of a complex covariance matrix (Greenhalgh et al., 2018). Figure A4 shows a polar
histogram (results of backazimuth calculations are grouped into regular bins of 30°). 15 clusters are pointing either towards
the back of the glacier (90-180°), where a crevasse is visible and where glacier is not frozen to the bed or the glacier front
(considering the 180° backazimuth ambiguity). The backazimuth is calculated from the north (0°).

The complex covariance matrix is formed over 0.05 s window of data to extract polarized seismic arrivals. The minimum
time-window length for the covariance matrix calculations should be at least half of the dominant period to ensure stable
backazimuth estimates (Greenhalgh et al., 2018). The 180° backazimuth ambiguity can be potentially resolved by determining
polarities of analysed arrivals. However, the time window of 0.05 s, depending on the signal wavelength, might incorporate
both positive and negative polarities making it impossible to reliably resolve 180° ambiguity. Reducing the time-window
length could help in assessing the polarity, but it would influence the stability of backazimuth measurements (i.e., the rank of
the covariance matrix).

An SVD of the covariance matrix provides eigenvector and eigenvalues describing the level of linear polarization and prop-
erties of polarized arrivals. The real components of the principal eigenvector are used to estimate the polarization direction. To
determine the level of linear polarization, we use a ratio between the first eigenvalue and the sum of the second and the third
eigenvalues. Assuming that seismic phases in the analyzed time window are strongly polarized, the covariance matrix will then
have a single dominant eigenvalue.

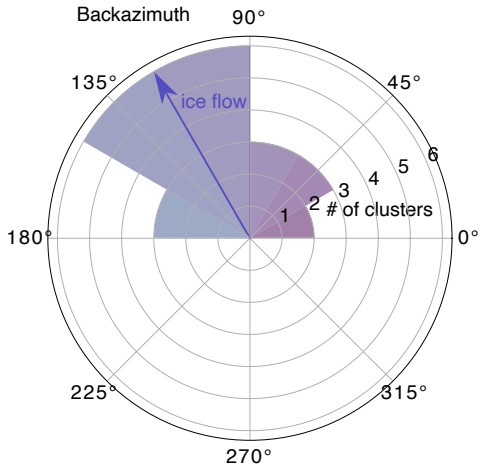

**Figure A4.** Polar histogram showing a distribution of the calculated backazimuth for different clusters.





**A4    Coda-wave interferometry and attenuation measurements**

Usually, icequakes are characterized by limited coda caused by lack of scatterers in glacier ice. However, highly damaged ice and the geometry of the Eiger hanging glacier generate strong icequake coda (Figure A5). This allows us to use the englacial coda to measure relative velocity variations ($dv/v$) and coda attenuation ($Q_c$) using the following processing scheme:

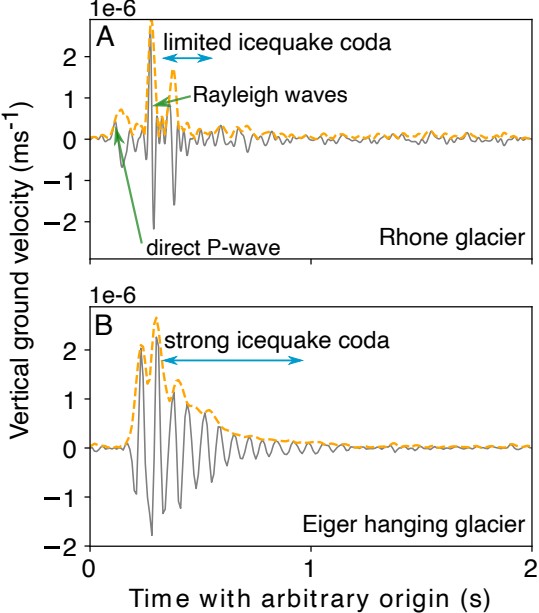

**Figure A5.** Icequake seismograms from (A) Rhone glacier and (B) Eiger hanging glacier (a single event from cluster 2). On the Rhone glacier we are able to identify different seismic phases (e.g., direct P-waves and Rayleigh waves), although on the Eiger hanging glacier due to the form/shape of the glacier and persisting scattering the seismic phases are mixed together which hinders identification of individual seismic arrivals.

1. Time-window duration. Coda wave interferometry (CWI) uses later times of seismograms in which the waves are suf-
ficiently scattered to contain waves travelling at many different directions (coda waves). We evaluate a lapse-time de-
       pendence of coda decay rate, to find out at which time noise becomes stronger than the scattered energy on retrieved
       icequake seismograms. After visual inspection of the decay of the envelope of the icequake seismograms, we chose the
       time window used in CWI as $T_{max}$= 1.5 s and $T_{min}$=0.5 s to avoid the influence from ballistic waves (Figure A6A-B).

2. Stability analysis. We analyze the temporal stability of icequake seismograms by correlating a stack of all icequake
seismograms (reference) and seismograms stacked in a given time window. We first stack icequake waveforms in regular
       bins of 1 h. Then, we evaluate the cross-correlation coefficient between the reference and the stack of icequakes using
       a running average over 1 h, 6 h, 12 h, 24 h, 48 h, and 3 days long time windows, with a step of 1 h (Figure A6). If the
       the stack is equivalent to the reference trace, the cross-correlation coefficient should be equal to 1. The 3-day stack

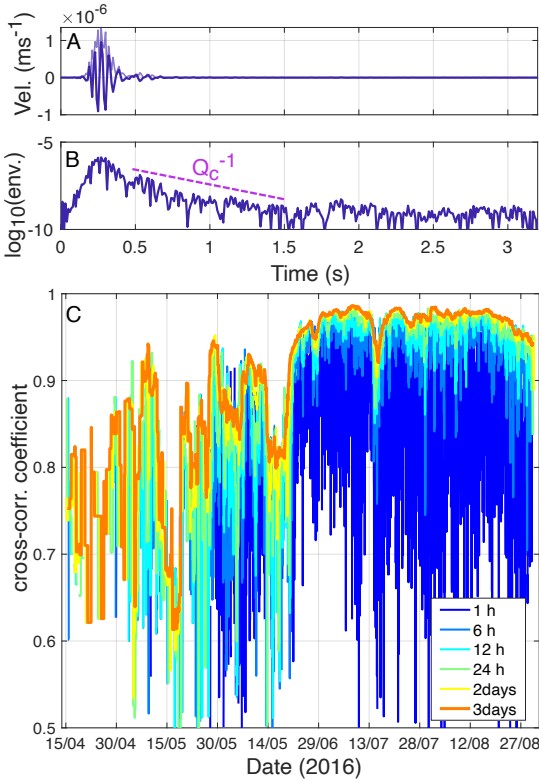

**Figure A6.** (A) Reference function in CWI: stack of all icequakes recorded at the vertical component of the EIG2 station for cluster 0, in the frequency-band (10-40) Hz and its envelope. (B) Lapse-time dependence of the envelope decay. (C) Cross-correlation coefficient as a function of stacking window duration.

of icequakes stabilizes the cross-correlation coefficient at >0.7. We then keep only 3-day stacks with cross-correlation

coefficients >0.7. The new reference is recalculated stacking only the selected icequakes. We average the icequakes over a 3-day moving time window with a step of 1 h to assess seasonal relative seismic velocity changes, and we use a 4-hour time window to monitor diurnal cycles.

3. Doublet method. We use the doublet method, also called Moving Window Cross Spectral technique (Fréchet et al., 1989; Clarke et al., 2011) to calculate $dv/v$. The doublet method operates in subsequent short (here: 3 times the maximum

period 0.1 s) sliding windows (overlap=80%) along the lag time. In each window, $dt$ is assumed to be constant and the current trace is considered to be a time-shifted version of the reference. For each segment, first the phase spectra difference is calculated (the phase of the crosspectrum for the reference and the current trace). A linear fit over the cross spectrum as a function of frequency provides the $dt$ value through the slope and error estimations. The measured $dt$ is assigned to the lapse time at the center of the time window ($t$). The time differences are calculated at a given time lag,

allowing us to assess the $dt/t$ value through the slope (Figure A7). Then, using the relationship for homogeneous velocity





change in a medium, (Snieder et al., 2019), we obtain: $dv/v = -dt/t$, where $dv/v$ is the relative velocity variation. The $dv/v$ error is calculated as the error of the ordinary least squares solution to the linear system of equations $t * dt/t = dt$. Figure A8A shows a spaghetti plot of all $dv/d$ measurements.

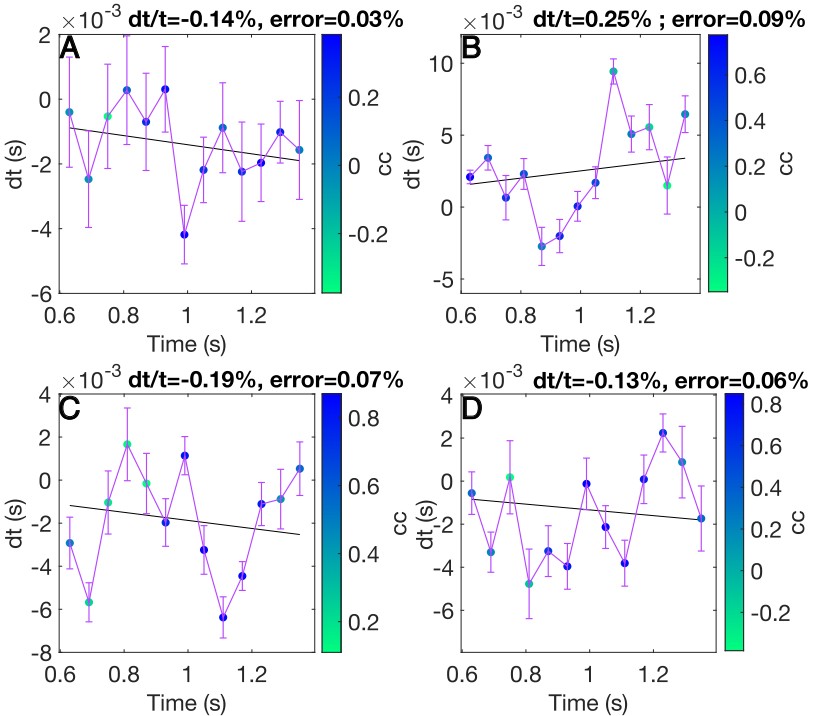

**Figure A7.** Example of $dt/t$ measurement using cross-correlation on moving windows and linear regression for cluster 0. Cross-correlation between the reference and analysed signals for each time windows is presented in color, $dt$ measurements in individual time windows with their error bars, at four dates: April 25, 2016, 20:00, (B) May 25, 2016, 15:00, (C) August 12, 2016, 19:00, and (D) August, 2016, 19:00.

4. Q estimation. To quantitatively characterize the envelope decay gradient, we use coda attenuation, on the basis of the model by Aki and Chouet (1975): $A(t, T_c) \propto t^{-n}\exp(-\frac{\pi t}{Q_c T_c})$, where $A$, $t$, and $T_c$ are envelope amplitude, lapse time and central period, and the power $n$ depends on a geometrical spreading. Multiplying a geometrical spreading factor to the left hand side and taking $log_{10}$, we estimate $Q_c$ using a linear regression analysis. In the model, we use the body wave geometrical spreading ($n$=1), although we also tested the surface wave geometrical spreading ($n$=2), and we found the same relative changes in the $Q_c$ with slightly lower $Q_c$ values (a median difference over the whole monitoring period between the two models $\Delta Q$=7). Figure A8B shows a spaghetti plot of all $Q_c$ measurements.

5. Diurnal cycles. To estimate diurnal cycles we use the same processing steps as in (3), although with a shorter temporal window=4 h and by stacking maximum 3 events per hour with the highest cross-correlation coefficient obtained from the template matching. We verified that the analysis with CWI with 72 h and 4 h time windows give coherent results. The





results of $Q_c$ measurements with 72 h and 4 h time windows agree within one standard deviation limit. We use only 3
events to limit the influence of variable number of events in the stack on the CWI results.

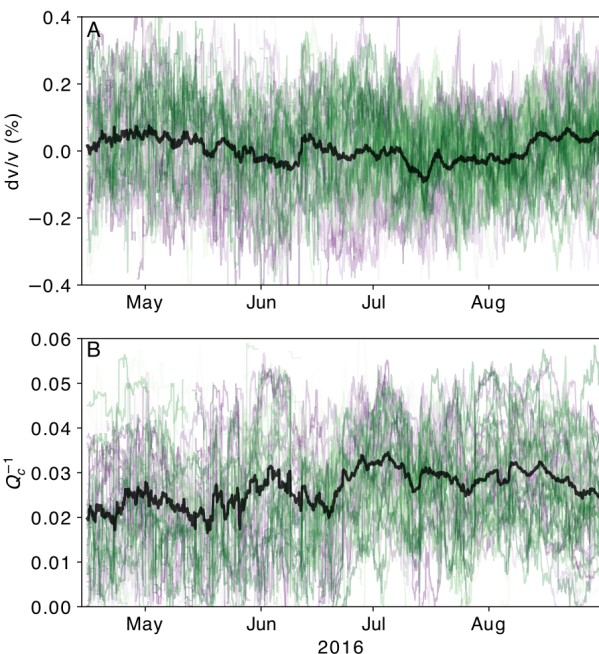

**Figure A8.** Spaghetti plots of $dv/v$ and $Q_c$ (no smoothing). Each line represents $dv/v$ and $Q_c$ measured over an individual cluster and
waveform components. The average over 69 measurements (23 clusterts x 3 components) are marked in black.

### A5    Sources of uncertainty

In CWI, we assume that the observed changes in coda wave arrival times are mostly related to changes in englacial seismic
velocities. However, they could be also related to changes in source-station distance, and changes in scattering properties of
the glacier. The GPS coordinates of the stations were measured on June 6, 2016 and August 31, 2016 (Preiswerk, 2018).
Station EIG2 moved around 1.4 m in between, which is equivalent to 16 mm per day, so 2.3 m approximately during the whole
monitoring period. Assuming that the position of sources is stable over the monitoring period, changes in station position
should be visible as a linear decreasing trend in out $dv/v$ measurements, although we do not observe it. This might be related
to the dominant wavelength $\lambda$ that we use ($\sim$65 m for surface waves) that is too large to be sensitive to a perturbation of the
order of 0.04 $\lambda$. This is also valid for the relative source relocalisation with CWI. If the sources originate from the glacier base,
they will change their position even less, as the surface glacier velocity integrates basal movement and elastic deformation of
the ice. The wavelength that we use would not allow us to measure properly such a small displacement. However, in the future,



with the use of higher frequencies, CWI could provide a tool to monitor source displacement in the glacier similarly to what has been done by Allstadt and Malone (2014).

Finally, we assume that the ice scattering properties relate mainly to large crevasses and do not change over the monitoring
period. The position of large crevasses can evolve with the glacier flow, although, as for the perturbation in source/receiver position, such perturbation driven by glacier flow would introduce waveform changes not reliably measurable with used dominant wavelengths.





## Appendix B: Figures

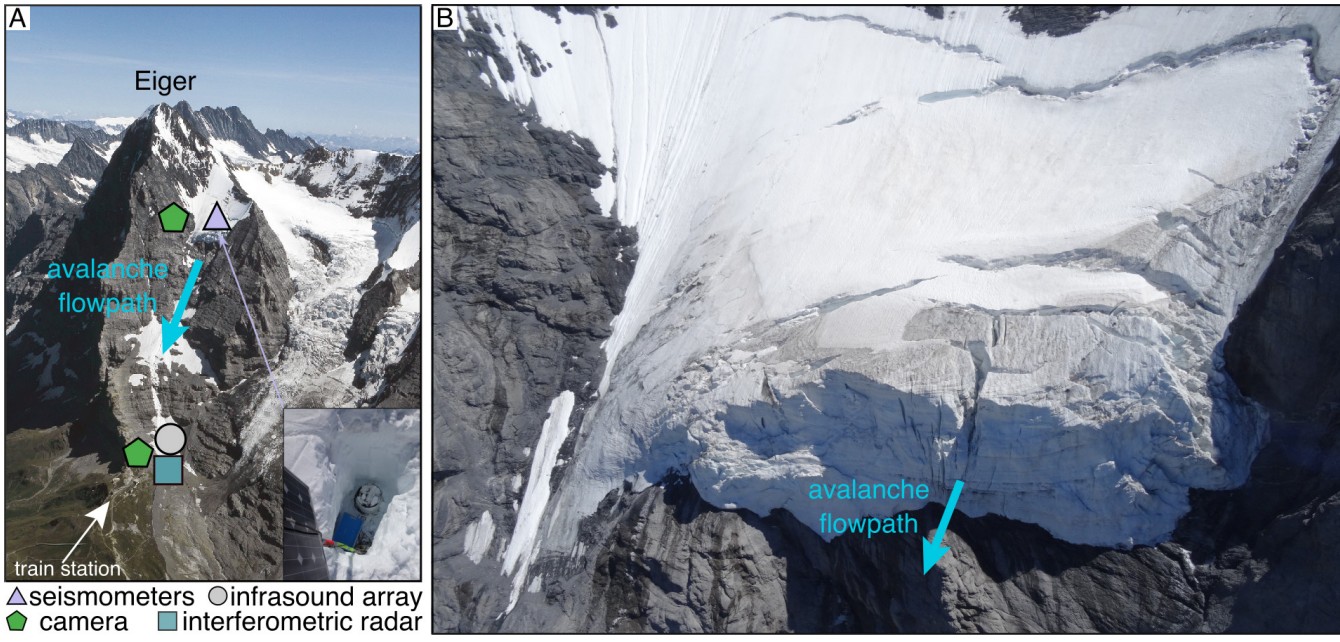

**Figure B1.** Overview over the Eiger hanging glacier. A. The following instruments were deployed on the glacier to monitor it: infrasound array (gray circle), interferometric radar measuring ice motion (teal square), 4 3-component seismometers (natural frequency: 1 Hz) installed on the glacier between April and August (stations EIG1:EIG4, triangles), an automatic camera photographing the unstable ice mass (green pentagons). The inset shows one of the seismic stations installed on a granite plate for an accurate leveling. The blue box contains the digitizer and battery. Source: ETH-Bibliothek Zürich, Bildarchiv /Stiftung Luftbild Schweiz/ Photograph: Swissair Photo AG / LBS_R2-010615 / CC BY-SA 4.0. B. The Eiger hanging glacier in August 2015.




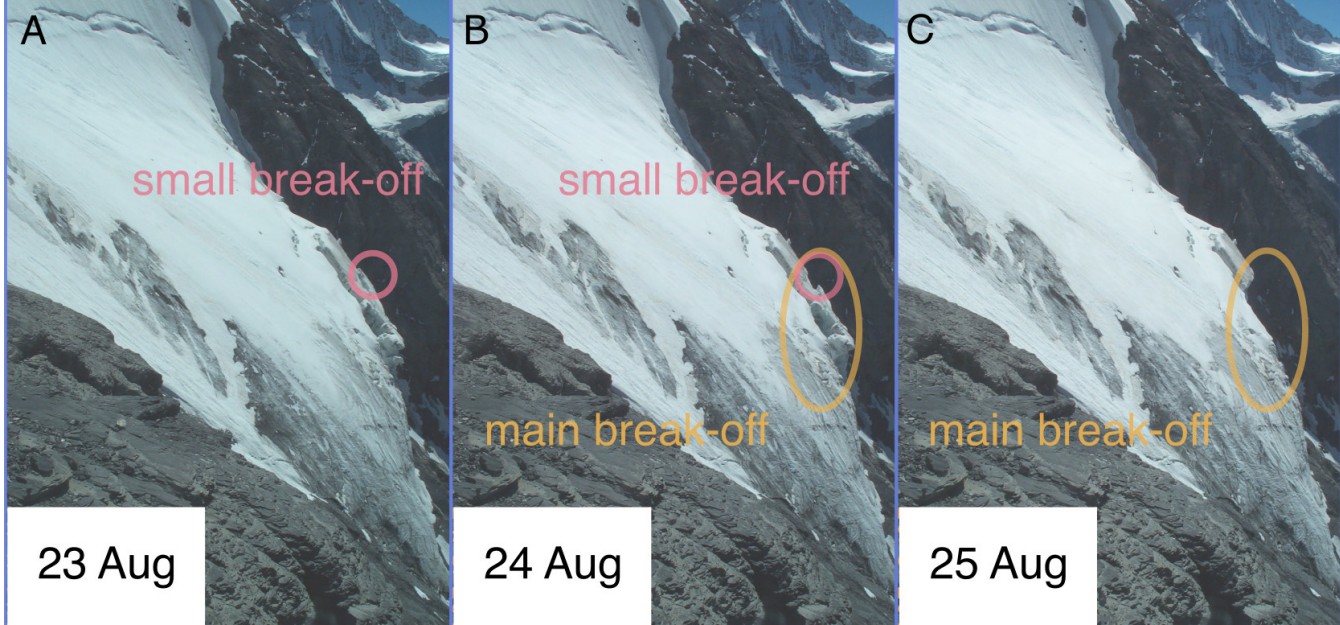

**Figure B2.** Lateral view of the glacier the automatic camera photographing the unstable ice mass. Photos from the automatic camera (A,B) before the small (23/08/16) and the main break-off event (24/08/16) correspondingly, and (C) after the break-off event (25/08/16).

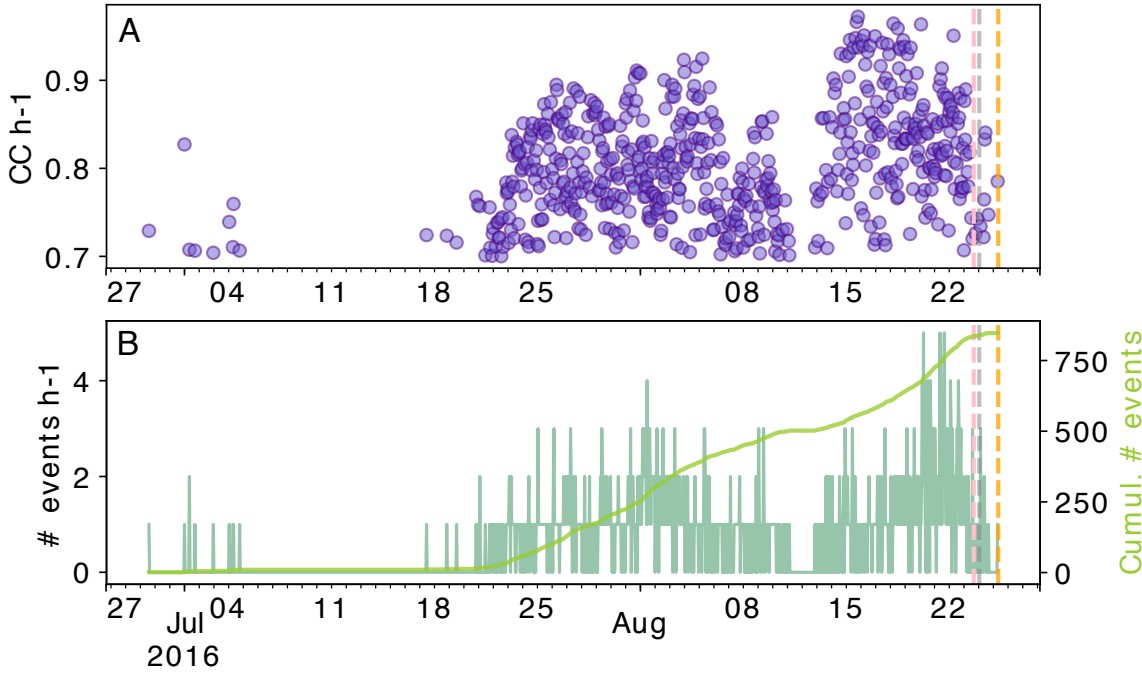

**Figure B3.** Activity of cluster 18. Only events with a correlation coefficient >0.7 are shown. (A) Hourly correlation coefficients. (B) Hourly detection rate (in green) with a cumulative number of events (in light green).

*Author contributions.* MC processed and analyzed the data with the help of FW. FW, LP, and MF installed the four seismometers and assured
data transmission. LM with Geopraevent installed the interferometric radar and provided the data. FB provided the $dv/v$ computation code.
MC prepared the manuscript with contributions from all co-authors.

*Competing interests.* The authors declare that they have no conflict of interest.

*Acknowledgements.* This work was funded by the Swiss National Science Foundation (SNSF) project Glacial Hazard Monitoring with
Seismology (GlaHMSeis, grant PP00P2 157551). We thank John Clinton, Roman Racine, and Stefan Wiemer; the Swiss Seismological
Service and its electronic laboratory (ELAB) for technical support and data archiving. We thank Geopraevent AG for allowing us to use their
high quality radar data and images. We thank Dominik Gräff, Amandine Sergeant, and Aurélien Mordret for helpful discussions.





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
