# Peer review of "Hanging glacier monitoring with icequake repeaters and seismic coda wave interferometry: a case study of the Eiger hanging glacier"

_Natural Hazards and Earth System Sciences, 2021_

## Author Comment (AC1)

Manuscript title: Hanging glacier monitoring with icequake repeaters and seismic coda wave interferometry: a case study of the Eiger hanging glacier

4 Manuscript ID: NHESS-2021-205

**6 Response to Reviewer 1**

**8 General comments**

This article proposes a seismological study of a hanging glacier in Swiss Alps, which has been
instrumented during five months in 2016. The authors recorded seismic data and used sophisticated methods to extract a lot of information from icequakes signals. Micro-seismicity
analysis and coda wave interferometry have been used by aiming at evaluating the temporal

- evolution of seismic indicators, such icequakes rate, relative seismic velocity and attenuation.
- 14 A major break-off event occurred at the end of the monitoring period, inviting the identification of precursors before the event for improving early warning systems.
- Although the technical challenge of seismic instrumentation, the main interest of the study lies in the investigation of physical processes in the subsurface of a hanging glacier, as a
   complementary method to forecasting technology focusing only the surface.
- I found that some results are not always very convincing (eg, back-azimuth from signal polarization) or some interpretations are rather speculative (basal slip?).
- Our response: We thank the Reviewer for this detailed and careful review. We performed
   some additional analysis to verify our results and to address the Reviewer's comments. We will
   also change the wording and rephrase our interpretations to make them less speculative
   following Reviewer's recommendations. We provide our detailed responses to each
- Reviewer's comment below. The Reviewer's comments are marked in blue, our responses in black, and parts of the manuscript are marked in *italic*.
- 28 I suggest several edits.

**Specific comments**

30 1) "repeating icequakes" ?

The term "repeating icequakes" or "repeaters" is present in many places in the manuscript,starting from the title. However, I don't think that this term is correct in this context, and I think it can be misleading.

Most "repeating icequakes" on glaciers have been detected at the base of glaciers or ice streams

- 2 (see references on 136-38). These events have both highly similar waveforms and quasiperiodic recurrence times. This suggests that they are associated with the repeating rupture of
- 4 asperities surrounded by aseismic slip. Repeating earthquakes are also defined as events having exactly the same rupture area, or at least an overlap of at least 50%.
- 6 See Uchida and Bürgmann (2019) for a review on repeaters in different contexts (faults, glaciers, landslides...).
- 8 In contrast, the events described in this study have similar waveforms but do not show any regularity in time. They have been detected using template matching with a correlation
- 10 threshold of 0.5, while a threshold of 0.9 is generally used to define "repeaters".
- This method thus groups together events that have similar waveforms in the frequency range
  10-40 Hz. This implies that events in the same cluster are likely separated by distances much smaller than this wavelength of about 65 m, but likely much larger than their rupture length.
- 14 There is no information on this study about the icequake magnitudes, so we cannot have an idea on the rupture length. The absence of regularity in time also suggests that icequake activity
- 16 is not associated with basal slip, but rather by crevasse opening.

I thus suggest to remove everywhere the term "repeating" or "repeaters" or to replace it by 18 "doublets", "multiplets" or "clusters", meaning events with similar waveforms.

- Our response: We thank the Reviewer for this comment and pointing out this inaccuracy of
   our manuscript. We agree that with the Reviewer that our method groups together icequakes
   that have similar waveforms, although due to the use low cross-correlations threshold of 0.5 in
- 22 the template matching, those events are "doublets" rather than "repeaters". We will change the title of the paper to: "Hanging glacier monitoring with icequake doublets and seismic coda
- 24 wave interferometry: a case study of the Eiger hanging glacier", and replace the term "repeaters" with "doublets" in the manuscript. We also thank the Reviewer for mentioning the
- 26 useful reference, we will add it to the manuscript. Also, we discuss the source separation between the two icequake doublets in below.
- 28 L90 : "The repeating events imply sources in close proximity with the same source mechanism, resulting in highly similar waveforms (Poupinet et al., 1984)"I don't agree with this statement,
- 30 such events are defined as "doublets" or "multiplets" (Poupinet et al., 1984).

In contrast, "Ideal repeaters represent two or more events that have exactly the same fault area and slip and thus produce the same seismic signal or waveform." (Uchida and Burgman, 2019).

34 **Our response:** As stated before, we thank the Reviewer for pointing out this inaccuracy in our manuscript. We will change the phrase to:

"Repeaters are two or more events that have exactly the same fault area and slip producing the same waveform with a threshold >0.9 (Uchida and Burgman, 2019). In our analysis, we first define templates by searching for icequake repeaters with RedPy (cross-correlation coefficient >0.9), and then we extend the analysis to icequake doublets through template matching (cross-correlations coefficient >0.5). Doublets are closely-spaced events, with

- 4 almost identical source mechanism, resulting in similar waveforms\_(Poupinet et al., 1984). Repeaters are two or more events that have exactly the same fault area and slip producing the
- 6 same waveform with a threshold >0.9 (Uchida and Burgman, 2019)."

2) Thermal regime and deformation mechanism

- 8 Eiger hanging glacier is a polythermal glacier but I don't understand which part of the base is cold.
- 10 L66: "The Eiger hanging glacier is polythermal [...], except the base of the frontal part which is cold (entirely frozen to the bed) (Lüthi and Funk, 1997)"
- 12 L164: "[...] the origin of most clusters either from the back of the glacier where a large crevasse is visible and where glacier is not frozen to the bed"
- These two sentences suggests that the front is cold, while the back is not frozen, that is a bit surprising. I don't understand German, so the reference (Lüthi and Funk, 1997) does not help
   me.
- Could you please clarify, and possibly indicate the transition between cold and temperate basal ice on a map?

How do you know the basal temperature, from boreholes to the base of the glacier ?

20 Could you also highlight the location of the crevasse mentioned on L164 on a map ?

Our response: The front of the glacier Eiger hanging glacier is cold, or at least it used to be in
 1993. Unfortunately, the only study that performed measurements on the Eiger hanging glacier
 dates back to 1997 (Lüthi and Funk, 1997) and the results of this study were only published in

- German. A more recent study by Magreth et al. (2017) presents some of the results of Lüthi and Funk, 1997 in English.
- 26 In a field campaign lasting several days in the spring of 1993, Lüthi and Funk determined the thickness of the glacier by using a ground penetrating radar along several longitudinal and
- 28 transverse profiles. In addition, seven boreholes were drilled in the glacier up to 70 m with hotwater drilling. The temperature of the glacier was determined using thermistors installed in the
- 30 boreholes. Finally, 15 stakes were drilled into the ice as reference points and their position was measured several times with interval of three weeks to determine the flow velocity on the
- 32 surface. Based on those measurements and using the finite element method, Lüthi and Funk provided a glacier model to determine the flow lines of the ice as well as the temperature and
- 34 stress distribution in the hanging glacier (Figure R1).

Their results show that coldest area is on the glacier bed close to the unstable front. This may
be surprising at first, but it might be explained by the glacier location. The shady location of the glacier front ensures low temperature and the ice cools down from the front. The rock

- 4 underneath is often snow-free and transfers the cold to the glacier bed. As a results, the ice in the glacier front is frozen to the ground, which contributes significantly to the stability of the
- 6 glacier.

We will add Figure R1 and the above explanations to the Appendix.

8

Figure R1: 2D section of Eiger hanging glacier. The cold glacier front (frozen to the glacier
bed) is shown on the left side. This ice flow and temperature model was obtained through the
finite element method. Englacial ice temperatures measured in three boreholes are also shown

- 12 (reprinted with a permission from Margreth et al., 2017, the original can be found in Luthi and Funk, 1997).
- 14

We will mark the crevasse in Figure 1 and B1, and B2.

16

3) Signal polarization and back-azimuth analysis

18 I am not totally convinced by the polarization analysis.

Could you illustrate the method by adding a figure showing a seismogram with arrival times of the different waves, and back-azimuth and linearity as a function of time? In several places you write that there is a  $180^{\circ}$  ambiguity in the estimated back-azimuth. I don't

- 2 understand why? Using the method of Vidale (BSSA, 1986), there is no ambiguity if the source is at depth.
- 4 Furthermore, you use a sliding time window of 0.05s to estimate the signal polarization, but I suspect that this time window is too large to separate P, S and surface waves. For illustrating,
- 6 Figure 1a suggests that seismic stations are located at about 50 m away from the crevasse. Assuming Vp=3600 m/s and Vs=1800 m/s, this gives a time delay of 0.014s between P and S
- 8 arrival times. There are thus likely both P, S and surface waves mixed in the same time window, with different polarizations. Also, there is a very strong coda, starting just after the first arrival
- 10 (Fig. A5B), with waves coming from different directions.

Moreover, why don't you also estimate the dip angle of P waves (corrected from free surface effect), in order to give an idea of the icequake depth?

L333: "The complex covariance matrix is formed over 0.05 s window of data to extract polarized seismic arrivals."

Can you specify "polarized"? Do you mean "linearly polarized"? What is the threshold you use for the linearity coefficient?

**Our response:** We thank the reviewer for these suggestions. We updated our polarization analysis and we follow exactly the steps presented by Vidale et al., 1986:

- We first calculate the covariance matrix and the we perform a singular value decomposition.
- 2) The eigenvector ( $x_0, y_0, z_0$ ) associated with the largest eigenvalue  $\lambda_1$  points in the direction of the largest amount of polarization. However, the phase in the complex plane of the eigenvectors is initially arbitrary.
- 1) The eigenvector ( $x_0, y_0, z_0$ ) is normalized to have length 1.
- 2) Then, we look for an angle α that maximizes the real component of the eigenvector associated with the largest eigenvalue. The angle α can be found by rotating the eigenvector by 0° to 180° in the complex plane. This rotation may be found by searching over α = 0° to 180° to maximize X, where X:

$$X = \sqrt{(\operatorname{Re}(x_0 \operatorname{cis} \alpha))^2 + (\operatorname{Re}(y_0 \operatorname{cis} \alpha))^2 + (\operatorname{Re}(z_0 \operatorname{cis} \alpha))^2} \quad (\text{Eq. 1})$$

30 Where:  $\operatorname{Re}(x)$  is the real part of x and  $\operatorname{cis}\alpha = \cos\alpha + i\sin\alpha$

3) The  $(x_0, y_0, z_0)$  is rotated by the angle alpha.

We then calculate the elliptical component of polarization (*Pe*) as a ratio of the imaginary part of the eigenvector to the real part of the eigenvector. We represent linearity as *Pl*=1-*Pe*:

$$P_l = 1 - P_e = 1 - \frac{\sqrt{1 - X^2}}{X}$$
 (Eq. 2)

2

4

8

10

5) The eigenvectors that correspond to the intermediate eigenvalue  $\lambda_2$  and smallest eigenvalue  $\lambda_3$  point in the directions of the intermediate and least amount of polarization, respectively. We use them to find the strength of the polarization signal *Ps*:

$$P_s = 1 - \frac{\lambda_2 + \lambda_3}{\lambda_1}$$
 (Eq. 3)

6 6) We determine the backazimuth of maximum polarization as (clockwise from the North in the range from 0° to 180°):

$$\phi = \frac{\pi}{2} + \arctan\frac{\operatorname{Re}(y_0)}{\operatorname{Re}(x_0)}$$
(Eq. 4)

7) And the dip of the direction of maximum polarization (clockwise from the vertical direction in the range from 0° to 180):

$$\delta = \frac{\pi}{2} + \arctan \frac{\operatorname{Re}(z_0)}{\sqrt{\operatorname{Re}(x_0)^2 + \operatorname{Re}(y_0)^2}}$$
(Eq. 5)

"The polarization vector is ambiguous in that the vector (x, y, z) represents the same polarization state as the vector (-x, -y, -z). In this paper, the strike and dip defined in equations (6) and (7) range from -90° to 90°, where 0° strike and dip represents a vector which points

16 horizontally in the direction back to the epicenter. The strikes in the range -180° to -90° and in the range 90° to 180° do not appear because of this ambiguity."

18 Regarding the duration of the sliding time window: We need to ensure a full rank covariance matrix to perform a singular value decomposition, meaning that the minimum duration of the 20 sliding time windows should be 3 samples. The data we use is sampled at 100 Hz, therefore

the minimum sliding time window duration is 0.03s. We initially decided to choose a longer
time window (0.05s) to provide a better estimate of the covariance matrix, although we agree
that such window duration potentially results in inclusion of two (or more) separate events.

24 Therefore, we updated the length of the time window used for the polarization analysis and recalculated the results. Also, we perform the polarization analysis in the frequency band of

26 (10-40) Hz (initially we did it on the raw data).

We thank the Reviewer for pointing out the time difference in individual arrivals of different
seismic phase (i.e., P- and S-waves, Rayleigh waves). The frequency sampling of 100 Hz might
be insufficient to distinguish relatively in between the arrivals of P and S waves. We will add
the following sentence to the manuscript:

"The low frequency sampling (Fs=100Hz) might further complicate the polarization analysis
since, the P and S wave arrivals might be separated by less than 0.3 s (e.g., assuming distance to the glacier front 50 m, Vp=3600 m/s and Vs=1800 m/s, this gives a time delay of 0.014s

34 between P and S arrival times, and 0.017 s between P and Rayleigh waves, and 0.004s between

the S and Rayleigh waves). It is though possible that certain sliding time windows might mix *P*-, *S*-, and surface waves."

2

4

We also estimate the dip as indicated in 7) although the distinct geometry of the glacier bed (Figure R2) complicates the interpretation of the dip values. We refrain from estimating the P-

- wave dip angle because the near-surface velocity profile of the glacier is poorly constrained.
  Estimating the P-wave dip angle requires knowledge of the velocity gradient, which we cannot provide at this moment. In contract to ablation zones where an equivalent technique has been
- 8 applied (Helmstetter et al., 2015) our glacier's near surface is characterized by snow/firn compaction and thus highly depth dependent.
- 10 Please find below the figure showing seismic waveforms recorded over Z, N, and E components, linearity, the strength of the polarization, estimated backazimuth, and dip as a function of time.

---

## Author Comment (AC2)

**Manuscript title:** Hanging glacier monitoring with icequake repeaters and seismic coda wave interferometry: a case study of the Eiger hanging glacier

Manuscript ID: NHESS-2021-205

**Response to Reviewer 2:**

Review of "Hanging glacier monitoring with icequake repeaters and seismic coda wave interferometry: a case study of the Eiger hanging glacier" by Chmiel et al.

This is a paper examining seismic data collected on a hanging glacier during a time period containing a break-off event from the front of the glacier in 2016. The seismic data are interpreted along with temperature data, surface velocity of the glacier front, and remote cameras. I think the paper is worthy of publication and came away with these comments:

**Our response:** We thank the Reviewer for this helpful review and the important comments. Please find our responses to Reviewer's comments below. We provide our detailed responses to each Reviewer's comment below. The Reviewer's comments are marked in blue, our responses in black, and parts of the manuscript are marked in *italic*.

- Figure B1 shows an infrasound array, but there is no mention of those data in the paper. Has there been analysis of the infrasound data and did it show anything besides presumably the signal from the large break-off event?

**Our response:** Unfortunately, the infrasound array was not operational during the break off event. However, a recent study by Marchetti et al., (2021) shows the analysis of the infrasound signals from a break-off event that occurred at the Eiger hanging glacier on May 29, 2017, and the potential for infrasound records to provide quantitative information of glacier collapse and ice avalanche trajectories, and possibly, volume. We will add the following sentence to the manuscript:

- The authors mention at line 80 that up to 3 of the seismic stations operated simultaneously at times. Which made me wonder if any array method could be applied to the 3 stations during that time, using the 3 stations as a tripartite array? Such beamforming (like what is done with infrasound arrays) could complement the polarization analysis.

**Our response:** We thank the Reviewer for this very useful suggestion. We performed an additional beamforming analysis to resolve the 180° ambiguity in the polarization analysis. Also, following the remarks of the Reviewer 1 we now present more details of the backazimuth analysis.

We use Matched Field Processing (MFP, Baggeroer et al. 1988; Kuperman and Turek 1997) to locate the icequakes. MPF exploits cross-array signal coherence and this method is based on the calculation of the cross-spectral correlation matrix (CSDM, Kuperman and Turek 1997), which represents the coherence of the wavefield recorded at a group of sensors. We do not go into technical details here, they can be found in Chmiel et al., 2019 and Bowden et al., 2021. We use both phase and amplitude information of the icequakes (Bowden et al., 2021) assuming a 2D medium with a Rayleigh wave velocity=1600 m/s.

The four stations were never in operation simultaneously, therefore for each cluster, and each station, we stack icequake waveforms recorded on the vertical component of the stations for time periods when the station was working properly. This provides us with average icequake waveform stacks for four stations. The value of the MFP is normalized in between 0 and 1 due to the normalization of the CSDM and model vector. We note though that the backazimuth estimates are obtained from the first arrivals, while MFP uses the entire icequake waveform (0-1.5s time window) assuming a 2-D Rayleigh wave propagation. This might cause a difference between the dominant backazimuth directions obtained from the two methods. We will add this analysis to the Appendix. Since the beamforming analysis is based on four stations it potentially yields more robust results compared to the single-station polarization analysis. In the revised manuscript, we will leverage the results of the two techniques to provide the most robust estimates of the backazimuth direction.

Please find below the result of the MFP for cluster 0:

[Figure]

Figure R1: MFP results for cluster 0 and the estimated backazimuth (including the 180°

ambiguity) with its uncertainty (marked in white dashed lines) obtained from the polarization
analysis. MFP allows us to resolve the 180° ambiguity in the backazimuth estimates.

*- In Appendix A5 the authors point out that the seismometers moved on the order of 1 meter during the deployment, which they argue does not affect their interpretation of the coda wave interferometry measurements. However, did the seismometers also happen to rotate at all in addition to the 1 meter of movement? Any rotation of the horizontal components could have an effect on the polarization analysis.*

**Our response:** We thank the Reviewer for this comment. We agree that the rotation of the seismometers would affect the polarization analysis. However, the stations remained horizontal on the granite plate with no significant rotation. The stations were snow covered: they were initially buried in the 1-2 m trenches and subsequently covered under ~1-2m of snow, which might have prevented them from turning. We will add this information to the manuscript.

*- Regarding the polarization analysis, was the same frequency bandpass used for it as was used for the coda wave interferometry (10-40 Hz)? What if there was significant frequency-dependency of the polarization over the band used? Have the authors looked at polarization in bandpassed data (e.g., 10-20 Hz, 20-30 Hz, 30-40 Hz) to see if the polarization is consistent as a function of frequency?*

**Our response:** We initially performed the polarization analysis on the raw data. We verified that the results are mostly coherent for the raw data and (10-40) Hz filtered data. However, to keep the consistency between different analysis method, we will use the results from (10-40) Hz in the final manuscript.

Following the Reviewer suggestion, we investigated the polarization analysis in different frequency bands: (10-40), (10-20) Hz, (20-30) Hz, (30-40) Hz. In the final manuscript, we will show the results in (10-40) Hz for the consistency with the beamforming analysis.

Changes for different frequencies can be related to the dominance of different seismic phases in different frequency bands (surface waves over body waves), and changing sensitivity of analysis due to a use of a fixed time window. However, by consequently choosing time windows for which the polarization strength is > 0.9 and that correspond to the highest linear polarization, our analysis yields coherent results for cluster 0 (Table 1 and Figure R2).

Finally, we will calculate the final backazimuth values as medians over frequency bands: (10-40), (10-20), (20-30), (30-40) Hz and we will provide uncertainty of the backazimuth estimates taken as a standard deviation of the estimated backazimuth values in four different frequency bands. We will add this information to the appendix.

[Figure]

Figure R2: Polarization analysis for cluster 0 (limited to the time window containing the first arrivals) in four different frequency bands: (10-40), (10-20) Hz, (20-30) Hz, (30-40) Hz. A. Raw waveforms recorded over Z, N, and E, component. B. linearity, C. the strength of the polarization, D. estimated backazimuth, and E. dip as a function of time.

|  | Backazimuth(°) | Dip(°) |
|---|---|---|
| (10-40) Hz | 48 | 56 |
| (10-20) Hz | 16 | 76 |
| (20-30) Hz | 22 | 66 |
| (30-40) Hz | 34 | 53 |
| Mean | 30 | 63 |
| Median | 28 | 61 |
| Std | 14 | 10 |
| Raw | 45 | 86 |

Table 1: Backazimuth and dip estimates in different frequency bands for cluster 0.

- I hate to say it, but I wasn't that impressed by the amount of fit in the dt/t plots in Figure A7. I normally like to see much better of a linear fit in this type of plot. Are the ones shown in this
figure typical? What could be causing the significant lack of a linear trend in these plots? Have the authors tried different approaches to defining the reference event? I wonder if there could
be an improvement by not even having a reference event and just measuring dt/t between all the events and inverting for a continuous function of dv/v, as was done by Hotovec-Ellis et al.
(2014, JGR; 2015, JGR). I think that approach is sometimes referred to as the "all doublet" method.

**Our response:** The dt/t plots shown in Figure A7 are rather typical in our analysis. We agree with the Reviewer that the fit is not impressive, although, it must be noted that these
measurements are made on a glacier, where scattering is still strongly limited compared to volcanic settings or crustal settings where CWI is commonly used. Moreover, the lack of linear
trend can be related to a localized perturbation in the glacier. Finally, the monochromatic coda visible for certain icequakes (Figures A3 and A5 in the manuscript) and source displacement
of 0.3 $\lambda$-0.4 $\lambda$ (see the comments of the Reviewer 1) can cause cycle skipping that would be visible as deviations in individual dt measurements. However, averaging dv/v measurements
over 69 individual measurements makes it statistically more representative and makes the estimation of final dv/v variations robust.

And a small correction: Figure A7 in the manuscript showed the results for the dt/t measurement for cluster 2, component E, not for cluster 0, component Z as stated in the
manuscript. We will correct the Figure caption in the new revised manuscript.

We performed different tests to verify our results:

1) We increased the overlap from 80% to 96% in measured time windows to add more measurement point in the linear regression (Figure R3). We also reject individual dt
measurements if dt>0.005s or error of dt> 0.005 s.

[Figure]

Figure R3: Examples of dt/t measurement using cross-correlation on sliding time windows and linear regression for cluster 2, component E. Cross-correlation between the reference and analysed signals for each time windows is presented in color, dt measurements in individual time windows with their error bars, at four dates: April 25, 2016, 20:00, (B) May 25, 2016, 15:00, (C) August 12, 2016, 19:00, and (D) August, 2016, 19:00 (with an overlap of 96%).

2) We adjusted the coda time window for 0.3-1s with an overlap of 96% (Figure R4). We also reject individual dt measurements if dt>0.005s or error of dt> 0.005 s. This gives a better fit of the induvial dt measurements.

[Figure]

Figure R4 Examples of dt/t measurement using cross-correlation on sliding time windows and linear regression for cluster 2, component E using coda in 0.3-1s time window. Cross-correlation between the reference and analysed signals for each time windows is presented in color, dt measurements in individual time windows with their error bars, at four dates: April 25, 2016, 20:00, (B) May 25, 2016, 15:00, (C) August 12, 2016, 19:00, and (D) August, 2016, 19:00 (with an overlap of 96%).

3) We tried also a moving reference approach as proposed by James et al., 2017. As a moving reference: we use the adjacent 3-day stack three days prior to the current day's stack. We also reject individual dt measurements if dt>0.005s or error of dt> 0.005 s. Then, to track the full velocity we cumulated the individual dv/v values. This approach does not significantly improve individual dt measurements and results in a linear drift caused by the error accumulation (James et al., 2017). We detrend the cumulative measurements to remove the linear drift due to the error accumulation. Note, that this might influence the accuracy of the results if there is a linear trend present in the dv/v measurements.

[Figure]

Figure R5: Examples of dt/t measurement using cross-correlation on sliding time windows and
linear regression for cluster 2, component E using a moving reference. Cross-correlation
between the reference and analysed signals for each time windows is presented in color, dt
measurements in individual time windows with their error bars, at four dates: April 25, 2016,
20:00, (B) May 25, 2016, 15:00, (C) August 12, 2016, 19:00, and (D) August, 2016, 19:00
(with an overlap of 96%). These measurements are not the same as measurements in Figure R3
and R4 due to the use of a moving reference, and the determined trends require a summation
to contribute to the overall dv/v measurements shown in R6.

4) We also test dv/v measurements using only icequakes that were detected in template
matching with a cross-correlation coefficient >0.7

The final results of the above tests show the same long-term variations as our initial results that
increases our confidence that the estimates of final dv/v shown in the manuscript were robust.
We will add a summary of this discussion and the references indicated by the Reviewer to the
Appendix.

[Figure]

Figure R6: Dv/v results from different approaches described above. The dv/v results obtained with a moving reference were scaled (dv/v/3) for the purpose of visual comparison.

- In research papers over the past decade, I don't often see the measurement of coda-Q but I appreciated it in this paper. How do the authors decide which portion of the event to measure
the Qc on as shown in Fig. A6B?

**Our response:** We thank the Reviewer for this comment. We chose the portion of the time
windows arbitrarily by choosing as the beginning the time where the envelope of the seismic signal starts to decay in a regular, linear manner (Aki and Chouet, 1975) and as the end, the
time where the envelope falls below the background noise levels (visible from 1.5s). We will add this information to the manuscript.

- The authors mention briefly that a period of increased seismicity correlated with the passage of a regional M6.2 earthquake in Fig. 1C. Have the authors looked in detail to see if increased
icequake activity is in fact triggered by the regional earthquake? Or is the increased event rate due to distant aftershocks that are not local?

**Our response:** Thank you for this comment. We looked in detail into the icequake activity.
The rate of icequake activity a few days before and after the earthquake is shown in Figure 1A in the manuscript and a few hours before and after the earthquake is shown in Figure R7 in this
rebuttal letter. In the revised manuscript, we will also add the rate of cluster activity to the panel C in Figure R7.

Since remote triggering usually occurs during the passage of teleseismic surface waves, not several hours later, we did not want to suggest that this burst of the seismic activity was
triggered by the Amartice earthquake. We will clarify this misunderstanding in the revised manuscript.

[Figure]

Figure R7: The rate of icequake activity for a few days before and after the M6.2 Amatrice earthquake. A. Vertical ground velocity recorded at the EIG2 station, B. the corresponding spectrogram, C. icequake activity between 21:00 August 23 and 08:00 August 24, 2016. The main shock and the aftershocks are shown in dashed gray lines (M6.2, M.5.5, M4.6, M4.3, M4.3, M4.4).

**Citation**: https://doi.org/10.5194/nhess-2021-205-RC2

**References**

Aki, K. and Chouet, B.: Origin of coda waves: source, attenuation, and scattering effects, Journal of geophysical research, 80, 3322–3342, 1975.

Baggeroer, A. B., Kuperman, W. A. and Schmidt, H., Matched field processing: Source localization in correlated noise as an optimum parameter estimation problem, J. of the Acoustical Soc. of Amer., vol. 83, no. 2, pp. 571-587, 1988.

Bowden, D.C, Sager, K., Fichtner, A., Chmiel, M., Connecting beamforming and kernel-based noise source inversion, Geophysical Journal International, 224, 3, 2021, 1607–1620, https://doi.org/10.1093/gji/ggaa539

Chmiel, M., Roux, P., Bardainne, T.; High-sensitivity microseismic monitoring: Automatic detection and localization of subsurface noise sources using matched-field processing and dense patch arrays. Geophysics 2019;; 84 (6): KS211–KS223. doi: https://doi.org/10.1190/geo2018-0537.1

James, S. R., H. A. Knox, R. E. Abbott, and E. J. Screaton (2017), Improved moving window cross-spectral analysis for resolving large temporal seismic velocity changes in permafrost, Geophys. Res. Lett., 44, 4018–4026, doi:10.1002/ 2016GL072468.

Kuperman, W.A., Turek, G., 1997. Matched field acoustics, Mech. Syst. Signal Process., 11, 141–148

Marchetti, E., Walter, F., & Meier, L. (2021). Broadband infrasound signal of a collapsing hanging glacier. Geophysical Research Letters, 48, e2021GL093579. https://doi.org/10.1029/2021GL093579